# Investigating pleiotropic effects of statins on ischemic heart disease in the UK Biobank using Mendelian randomisation

**CM Schooling[1,2]\*, JV Zhao[1], SL Au Yeung[1], GM Leung[1]**

[1]School of Public Health, Li Ka Shing Faculty of Medicine, The University of Hong Kong, Hong Kong, China; [2]City University of New York, Graduate School of Public Health and Health Policy, New York, United States

**Abstract** We examined whether specifically statins, of the major lipid modifiers (statins, proprotein convertase subtilisin/kexin type 9 (PCSK9) inhibitors and ezetimibe) have pleiotropic effects on ischemic heart disease (IHD) via testosterone in men or women. As a validation, we similarly assessed whether a drug that unexpectedly likely increases IHD also operates via testosterone. Using previously published genetic instruments we conducted a sex-specific univariable and multivariable Mendelian randomization study in the UK Biobank, including 179918 men with 25410 IHD cases and 212080 women with 12511 IHD cases. Of these three lipid modifiers, only genetically mimicking the effects of statins in men affected testosterone, which partly mediated effects on IHD. Correspondingly, genetically mimicking effects of anakinra on testosterone and IHD presented a reverse pattern to that for statins. These insights may facilitate the development of new interventions for cardiovascular diseases as well as highlighting the importance of sex-specific explanations, investigations, prevention and treatment.

**\*For correspondence:**
cms1@hku.hk

**Competing interests:** The authors declare that no competing interests exist.

## Introduction

Statins are the first-line lipid modifier for reducing cardiovascular morbidity and mortality (*Ray et al., 2019*; *Michos et al., 2019*). Statins have revolutionized the prevention and treatment of cardiovascular disease, and inspired the development of a range of effective interventions targeting the reduction of low-density lipoprotein (LDL)-cholesterol. Statins have long been suspected of having additional beneficial effects beyond lipid modulation (*Schonbeck and Libby, 2004*), such as on inflammation (*Schonbeck and Libby, 2004*), another potential target for reducing cardiovascular disease (*Aday and Ridker, 2018*). Meta-analysis of randomized controlled trials (RCTs) suggests statins are more effective at reducing mortality than proprotein convertase subtilisin/kexin type 9 (PCSK9) inhibitors or ezetimibe (*Schmidt et al., 2017*; *Khan et al., 2018*). However, these findings may be more apparent than real, stemming from differences in trial design, such as shorter duration of the PCSK9 inhibitor trials (*Khan et al., 2018*), the predominance of industry funded statin trials (*Hobbs et al., 2016*) or the difficulty of interpreting trials of 'soft' events when the treatment affects diagnostically relevant criteria, that is lipid levels (*Schooling and Zhao, 2019*). To investigate this anomaly, a previous study conducted a systematic agnostic scan of metabolic profile in a trial of statins compared to a PCSK9 inhibitor, which found few differences (*Sliz et al., 2018*). While characterization of the metabolic effects of statins suggested extensive effects on lipids and fatty acids (*Würtz et al., 2016*); these investigations were not able to include a factor which has previously been proposed as contributing to statin's effectiveness, that is effects on male hormones (*Schooling et al., 2013*), although questions have been raised as to whether statins are as effective in women as men (*Plakogiannis and Arif, 2016*). However, meta-analysis of the available trial evidence suggests similar relative benefits of LDL-cholesterol reduction by statins for men and women

(*Fulcher et al., 2015*), although the trials mainly concern men (73.2%) which may preclude detection of important sex differences. Men are also at substantially higher risk than women (*Ezzati et al., 2015*) giving larger absolute benefits in men than women at the same reduction in relative risk.

RCTs are not usually designed or powered to test mediating mechanisms. In addition, trials of statins on cardiovascular disease outcomes designed to be sex-specific are lacking. To assess a potential pathway by which statins might additionally operate, we used Mendelian randomization (MR), an observational study design that avoids confounding by taking advantage of the random allocation of genetic material at conception (*Smith and Ebrahim, 2003*), here specifically genetic variants mimicking effects of lipid modifiers. This random allocation at conception also avoids selection bias as long as few deaths have occurred between randomization and recruitment due to exposure, outcome, or other causes, that is competing risk, of the outcome (*Schooling et al., 2020*). So, here we focused on ischemic heart disease (IHD) (*Kesteloot and Decramer, 2008*), using the UK Biobank (*Collins, 2012*) to investigate whether testosterone mediated any of the effects of statins, PCSK9 inhibitors or ezetimibe on IHD in men or women using univariable and multivariable MR. As a further test, given some anti-inflammatories have also been shown to have opposite effects on male hormones compared to statins, specifically the interleukin one receptor antagonist (IL-1Ra), anakinra (*Ebrahimi et al., 2018*), we assessed whether the genetic variants mimicking effects of anakinra or tocilizumab targeting the interleukin six receptor (IL-6r) had opposite patterns of effects on testosterone and IHD (*Aday and Ridker, 2018*; *Swerdlow et al., 2012*; *Interleukin 1 Genetics Consortium, 2015*) to statins. *Figure 1* illustrates the possible additional effects of statins, anakinra or tocilizumab on IHD via male hormones in the context of the well-established benefits of statins, PCSK9 inhibitors and ezetimibe acting via LDL-cholesterol and of anti-inflammatories in IHD.

## Results

The six single neucleotide polymorphisms (SNPs) mimicking effects of statin (rs12916, rs5909, rs10066707, rs17238484, rs2006760 and rs2303152 from *HMGCR*) (*Ference et al., 2019*) were all correlated ($r^2$ >0.13). In the main analysis we used only the lead SNP, rs12916. Of the 7 SNPs mimicking effects of PCSK9 inhibitors (rs11206510, rs2149041, rs7552841, rs10888897, rs2479394, rs2479409 and rs562556 from *PCSK9*) (*Ference et al., 2019*), the three independently ($r^2$ <0.05) and most strongly associated with LDL-cholesterol (rs11206510, rs2149041 and rs7552841) were used in the main analysis. Of the 5 SNPs mimicking effects of ezetimibe (rs10260606 (proxy of rs2073547, $r^2$ = 0.99), rs2300414, rs10234070, rs7791240, rs217386 from *NCP1L1*) (*Ference et al., 2019*), two SNPs (rs2300414 and rs10234070) were discarded because their F-statistic for LDL-cholesterol was <10. The remaining three SNPs were all correlated at $r^2$ >0.05. rs2073547 was used in the main analysis because it had the strongest association with LDL-cholesterol. *Supplementary file 1a* shows

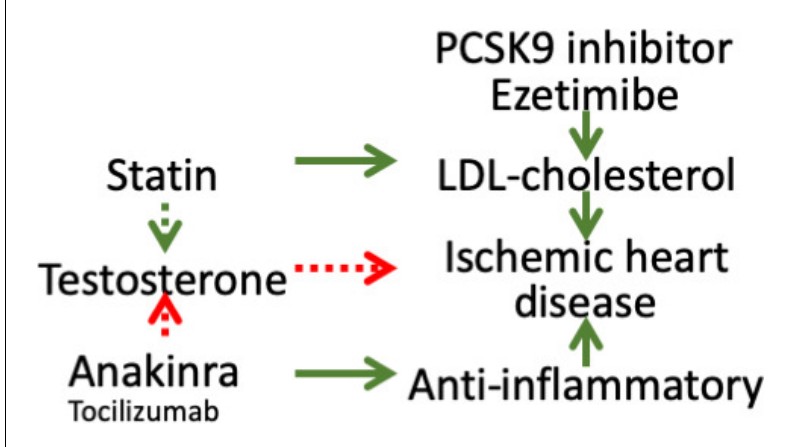

**Figure 1.** Directed acyclic graph showing the well-established protective effects of lipid modifiers and anti-inflammatories on IHD (solid lines) and possible additional pathways (dashed lines) investigated here.  Green indicates a lowering effect, red indicates an increasing effect.

the associations with LDL-cholesterol by sex for the independent SNPs used to mimic effects of statins, PCSK9 inhibitors and ezetimibe. *Supplementary file 1b* shows the associations of the two SNPs mimicking effects of the anti-inflammatory, anakinra, (rs6743376 and rs1542176 ($r^2$ <0.001)) with IL-1Ra and the associations of the SNP (rs7529229), mimicking effects of tocilizumab use, with IL-6r.

There were 179,918 men with 25,410 cases of IHD and 212,080 women with 12,511 cases of IHD in the UK Biobank.

## Instrument strength

The F-statistics for SNPs used to genetically mimic the effects of statins, PCSK9 inhibitors and ezetimibe were all >10 in men and women (*Supplementary file 1a*), as were the F-statistics for the SNPs used to mimic the effects of anakinra and tocilizumab (*Supplementary file 1b*). The F-statistics for the 125 and 254 SNPs predicting testosterone in men and women were all greater than 10, with mean 128.6 and 83.3, respectively.

## Sex-specific associations of genetically mimicked lipid modifiers with testosterone

Genetically mimicked effects of statins reduced testosterone in men but not women (*Table 1*). Genetically mimicked effects of PCSK9 inhibitors and of ezetimibe did not affect testosterone in men or women (*Table 1*). Findings were similar in sensitivity analysis including correlated SNPs,

**Table 1.** Sex-specific Mendelian randomization estimates (where possible) for effects of genetically mimicked statins, PCSK9 inhibitor and ezetimibe (in effect sizes of LDL-cholesterol) on testosterone (effect size) in men and women using the UK Biobank .

| | Therapy | # SNPs | Method | Beta | 95% CI | P value | MR-Egger intercept p-value |
|---|---|---|---|---|---|---|---|
| | | | | **Mendelian Randomization estimates** | | | |
| Men | Statin | 1 | Inverse variance weighted | −0.15 | −0.23 to −0.06 | 0.001 | |
| | Statin | 6 | Inverse variance weighted | −0.15 | −0.23 to −0.07 | 0.0005 | |
| | PCSK9 inhibitor | 3 | Inverse variance weighted | 0.04 | −0.11 to 0.18 | 0.63 | |
| | PCSK9 inhibitor | 3 | Weighted median | 0.07 | −0.13 to 0.27 | 0.29 | |
| | PCSK9 inhibitor | 3 | MR-Egger | 0.34 | 0.09 to 0.60 | 0.01 | −0.01 (0.01) |
| | PCSK9 inhibitor | 7 | Inverse variance weighted | 0.05 | −0.05 to 0.15 | 0.29 | |
| | ezetimibe | 1 | Inverse variance weighted | 0.04 | −0.15 to 0.23 | 0.68 | |
| | ezetimibe | 3 | Inverse variance weighted | 0.05 | −0.12 to 0.22 | 0.55 | |
| | ezetimibe | 3 | Weighted median | 0.03 | −0.13 to 0.18 | 0.72 | |
| | ezetimibe | 3 | MR-Egger | 0.24 | −0.52 to 1.0 | 0.54 | −0.01 (0.52) |
| Women | Statin | 1 | Inverse variance weighted | 0.04 | −0.06 to 0.14 | 0.45 | |
| | Statin | 6 | Inverse variance weighted | 0.03 | −0.07 to 0.13 | 0.52 | |
| | PCSK9 inhibitor | 3 | Inverse variance weighted | 0.01 | −0.11 to 0.14 | 0.85 | |
| | PCSK9 inhibitor | 3 | Weighted median | 0.01 | −0.13 to 0.15 | 0.91 | |
| | PCSK9 inhibitor | 3 | MR-Egger | 0.09 | −0.38 to 0.56 | 0.71 | −0.003 (0.74) |
| | PCSK9 inhibitor | 7 | Inverse variance weighted | −0.004 | −0.14 to 0.13 | 0.95 | |
| | ezetimibe | 1 | Inverse variance weighted | 0.18 | −0.05 to 0.40 | 0.12 | |
| | ezetimibe | 3 | Inverse variance weighted | 0.12 | −0.08 to 0.31 | 0.24 | |

One statin SNP is rs12916, and six statin SNPs additionally included rs5909, rs10066707, rs17238484, rs2006760 and rs2303152 taking into account their correlations.

Three PCSK9 inhibitor SNPs are rs11206510, rs2149041 and rs7552841, and 7 PCSK9 inhibitor SNPs additionally included rs10888897, rs2479394, rs2479409 and, rs562556 taking into account all their correlations.

One ezetimibe SNP is rs2073547 (proxied by rs10260606), and three ezetimibe SNPs additionally included rs7791240 and rs217386 taking into account all their correlations.

The unit of LDL-cholesterol is approximately 0.83 mm/L. An effect size of testosterone is approximately, 0.23 nmol/L in women (*Haring et al., 2012*) and 3.1 nmol/L in men (*Mohr et al., 2005*).

where available (**Table 1**). PCSK9 inhibitors and ezetimibe were not investigated further, given the lack of association with testosterone in men and women.

## Sex-specific associations of genetically mimicked statin use and testosterone with IHD

Genetically mimicked effects of statins reduced the risk of IHD in men and possibly women (**Table 2**) using IVW. Steiger filtering indicated directionality from testosterone to IHD in men and women. Genetically predicted testosterone was positively associated with IHD in men, but was not significantly associated with IHD in women, with similar estimates using IVW, the weighted median and MR-Egger. MR-Egger intercepts did not suggest the IVW estimates were invalid, but had wider confidence intervals (**Table 2**).

Considering genetically mimicked effects of statin together with genetically predicted testosterone, in men the multivariable estimates for genetically mimicked effects of statins on IHD allowing for testosterone we attenuated (**Table 3**) compared to the univariable estimates for effects of statins on IHD (**Table 2**). As a result, the multivariable MR-Egger estimates for genetically mimicked effects of statins on IHD, allowing for testosterone, were very similar for men and women (odds ratio 0.72, 95% confidence interval 0.57 to 0.90 for men and women meta-analyzed together). The multivariable associations of genetically predicted testosterone with IHD in men and women, allowing for genetically mimicked statins, (**Table 3**) were very similar to the respective univariable estimates for men and women (**Table 2**), but differed by sex (z-test p-value 0.01). The conditional F-statistics were 58.2 (men) and 68.5 (women) for testosterone and 3.5 (men) and 6.8 (women) for effects of genetically mimicked statins. The Q statistics for instrument validity were significant (212.5 in men and 323.1 in women), and the multivariable MR-Egger intercepts were significant in men and women, substantiating the use of the MR-Egger estimates.

## Sex-specific associations of genetically mimicked Anakinra and tocilizumab with testosterone and IHD

Genetically mimicked effects of anakinra increased both the risk of IHD and testosterone in men but not women (**Table 4**). Genetically mimicked effects of tocilizumab were not clearly associated with testosterone in men or women (**Table 4**), so were not investigated further. Investigation of whether

**Table 2.** Mendelian randomization estimates for effects of genetically mimicked statins (effect sizes of LDL-cholesterol) and of genetically predicted testosterone (effect size) on IHD in men and women using the UK Biobank.

| | Exposure | # SNPs | Method | Mendelian randomization estimates | | | |
| | | | | OR | 95% CI | P value | MR-Egger intercept p-value |
|---|---|---|---|---|---|---|---|
| Men | Statin mimic | 1 | Inverse variance weighted | 0.55 | 0.38 to 0.79 | 0.001 | |
| | Statin mimic | 6 | Inverse variance weighted | 0.54 | 0.33 to 0.89 | 0.02 | |
| | Testosterone | 125 | Inverse variance weighted | 1.11 | 1.04 to 1.19 | 0.003 | |
| | Testosterone | 125 | Weighted median | 1.18 | 1.06 to 1.31 | 0.002 | |
| | Testosterone | 125 | MR-Egger | 1.10 | 0.98 to 1.23 | 0.09 | 0.01 (0.84) |
| Women | Statin mimic | 1 | Inverse variance weighted | 0.87 | 0.59 to 1.27 | 0.46 | |
| | Statin mimic | 6 | Inverse variance weighted | 0.79 | 0.54 to 1.13 | 0.20 | |
| | Testosterone | 254 | Inverse variance weighted | 0.96 | 0.89 to 1.03 | 0.29 | |
| | Testosterone | 254 | Weighted median | 1.03 | 0.92 to 1.14 | 0.63 | |
| | Testosterone | 254 | MR-Egger | 1.08 | 0.94 to 1.23 | 0.27 | −0.004 (0.05) |

One statin SNP is rs12916, and six statin SNPs additionally included rs5909, rs10066707, rs17238484, rs2006760 and rs2303152 taking into account all their correlations. The unit of LDL-cholesterol is approximately 0.83 mm/L. An effect size of testosterone is approximately, 0.23 nmol/L in women (**Haring et al., 2012**) and 3.1 nmol/L in men (**Mohr et al., 2005**).

The online version of this article includes the following source data for Table 2:

**Source data 1.** Genetic associations for men.

**Source data 2.** genetic associations for women.

**Table 3.** Multivariable Mendelian randomization estimates for effects of genetically mimicked statins (effect sizes of LDL-cholesterol) and of testosterone (effect size) together on IHD in men and women using the UK Biobank.

| Sex | Exposures | Instrumented by | Adjusted for | Method | OR | 95% CI | P value | MR-Egger intercept p-value |
|---|---|---|---|---|---|---|---|---|
| Men | Statin mimic | 1 Statin SNP on LDL-cholesterol | Testosterone | Inverse variance weighted | 1.05 | 0.74 to 1.47 | 0.79 | |
| | Testosterone | 125 SNPs on testosterone | statin | Inverse variance weighted | 1.11 | 1.04 to 1.20 | 0.003 | |
| | Statin mimic | 1 Statin SNP on LDL-cholesterol | Testosterone | MR-Egger | 0.73 | 0.48 to 1.11 | 0.14 | |
| | Testosterone | 125 SNPs on testosterone | statin | MR-Egger | 1.09 | 1.02 to 1.17 | 0.02 | 0.005 |
| | Statin mimic | 6 Statin SNPs on LDL-cholesterol | Testosterone | Inverse variance weighted | 1.02 | 0.72 to 1.43 | 0.91 | |
| | Testosterone | 125 SNPs on testosterone | statin | Inverse variance weighted | 1.11 | 1.04 to 1.20 | 0.003 | |
| Women | Statin mimic | 1 Statin SNP on LDL-cholesterol | Testosterone | Inverse variance weighted | 0.98 | 0.75 to 1.16 | 0.53 | |
| | Testosterone | 254 SNPs on testosterone | statin | Inverse variance weighted | 0.96 | 0.90 to 1.04 | 0.33 | |
| | Statin mimic | 1 Statin SNP on LDL-cholesterol | Testosterone | MR-Egger | 0.72 | 0.55 to 0.94 | 0.02 | |
| | Testosterone | 254 SNPs on testosterone | statin | MR-Egger | 0.96 | 0.89 to 1.03 | 0.27 | 0.001 |
| | Statin mimic | 6 Statin SNPs on LDL-cholesterol | Testosterone | Inverse variance weighted | 0.92 | 0.74 to 1.16 | 0.49 | |
| | Testosterone | 254 SNPs on testosterone | statin | Inverse variance weighted | 0.97 | 0.90 to 1.04 | 0.36 | |

One statin SNP is rs12916, and six statin SNPs additionally included rs5909, rs10066707, rs17238484, rs2006760 and rs2303152 taking into account all their correlations. The unit of LDL-cholesterol is approximately 0.83 mm/L. An effect size of testosterone is approximately, 0.23 nmol/L in women (**Haring et al., 2012**) and 3.1 nmol/L in men (**Mohr et al., 2005**).

testosterone mediates the genetically mimicked effect of anakinra on IHD was not possible because sex-specific genetic associations of testosterone SNPs with IL-1Ra from suitably large GWAS are not available.

**Table 4.** Mendelian randomization inverse variance weighted estimates for genetically mimicked effects of the anti-inflammatory anakinra raising IL-1Ra (effect size) (**Swerdlow et al., 2012**) on testosterone (effect size) and ischemic heart disease and for genetically mimicked effects of tocilizumab raising serum IL-6r (ng/ml) (**Rafiq et al., 2007**) on testosterone in men and women using the UK Biobank .

| | Therapy | Target | Outcome | # SNPs | Measure | Estimate | 95% CI | p-value |
|---|---|---|---|---|---|---|---|---|
| Men | Anakinra | IL-1Ra | testosterone | 2 | beta | 0.022 | 0.01 to 0.04 | 0.002 |
| | | | IHD | 2 | OR | 1.08 | 1.01 to 1.15 | 0.017 |
| | Tocilizumab | IL-6r | testosterone | 1 | beta | 0.003 | −0.06 to 0.13 | 0.96 |
| Women | Anakinra | IL-1Ra | testosterone | 2 | beta | −0.01 | −0.04 to 0.01 | 0.24 |
| | | | IHD | 2 | OR | 0.99 | 0.91 to 1.08 | 0.86 |
| | Tocilizumab | IL-6r | testosterone | 1 | beta | 0.002 | −0.02 to 0.02 | 0.84 |

SNPs mimicking anakinra are rs6743376 and rs1542176.
The SNP mimicking tocilizumab is rs7529229.
An effect size of testosterone is approximately, 0.23 nmol/L in women (**Haring et al., 2012**) and 3.1 nmol/L in men (**Mohr et al., 2005**).

## Discussion

Consistent with meta-analysis of RCTs this study provides genetic evidence that statins reduce testosterone in men (*Schooling et al., 2013*), and adds by showing that statins could partially operate on IHD, in men only, by reducing testosterone, as previously hypothesized (*Schooling et al., 2013*; *Schooling et al., 2014*) while having similar protective effects in men and women independent of testosterone. Previous Mendelian randomization studies have shown lower testosterone associated with lower risk of IHD, particularly in men (*Schooling et al., 2018a*; *Luo et al., 2019*; *Mohammadi-Shemirani et al., 2019*). Conversely, consistent with an RCT (*Ebrahimi et al., 2018*), this study also provides genetic validation that the anti-inflammatory anakinra, targeting IL-1Ra, increases testosterone in men, and is consistent with a previous Mendelian randomization study showing anakinra increases IHD (*Interleukin 1 Genetics Consortium, 2015*), but adds by showing why these associations might occur and that they may be specific to men.

A more marked association of testosterone with IHD in men than women (*Table 2*) is consistent with sex differences in biology, where testosterone is the main sex hormone in men and is much higher in men than in women. The associations of genetically mimicked effects of statins, PCSK9 inhibitors or ezetimibe on testosterone is consistent with the evidence available (*Schooling et al., 2013*; *Ooi et al., 2015*; *Krysiak et al., 2015*) and their mechanisms of action. Specifically, statins inhibit cholesterol synthesis, while PCSK9 inhibitors enable greater clearance of cholesterol, through increasing LDL-receptors, while ezetimibe reduces uptake of dietary cholesterol (*Ray et al., 2019*; *Michos et al., 2019*). However, some cells, such as Leydig cells, use de novo cholesterol synthesis to generate steroids, which can be reduced by statins (*Shimizu-Albergine et al., 2016*). Concerns about statins compromising androgen production pre-date the marketing of statins (*Farnsworth et al., 1987*; *MacDonald et al., 1988*). Beneficial immunosuppressive effects of androgens in rheumatoid arthritis have long been known (*Cutolo et al., 1991*), making androgen reduction a plausible mode of action for therapies, such as anakinra, whose primary indication is rheumatoid arthritis.

These findings may seem counter-intuitive given the essential role of testosterone in masculinity and reproduction. However, in 2015 the Food and Drug Administration in the United States required labelling changes for all testosterone prescriptions to warn of the risk of heart attacks and stroke on testosterone, although no sufficiently large RCT of testosterone administration has been conducted to confirm these effects (*Onasanya et al., 2016*). The Endocrine Society has also recommended caution in the use of testosterone (*Bhasin et al., 2018*). Meta-analysis of RCTs suggests androgen deprivation therapy reduces all-cause mortality, but is too small to quantify effects on specific diseases beyond prostate cancer (*Nguyen et al., 2011*). As such, our Mendelian randomization findings of the effects of testosterone have some consistency with the limited experimental evidence. In addition, our findings are consistent with well-established evolutionary biology theory, that is reproductive success may be at the expense of longevity, possibly in a sex-specific manner, impling that central drivers of the reproductive axis, as well as androgen production and catabolism, and their environmental cues may be relevant to IHD (*Schooling, 2016*; *Schooling and Ng, 2019*; *Figure 2*) encompassing the relations tested here (*Figure 1*). Notably, upregulation of indicators of plentiful living conditions, such as insulin, appear to cause IHD, particularly in men (*Zhao et al., 2019*), likely via gonadotropin releasing hormone (GnRH) (*Schooling and Ng, 2019*). Similarly, fatty acids may affect GnRH (*Tran et al., 2016*; *Matsuyama and Kimura, 2015*). In contrast, indicators of adversity, such as endotoxins promote an inflammatory response, involving interleukins, which suppresses the reproductive axis (*Kalra et al., 1998*) and thereby testosterone (*Tremellen et al., 2018*), which may be reversed by anakinra possibly outweighing the benefits for IHD of suppressing inflammation (*Tardif et al., 2019*). Statins, in contrast reduce androgen production, while agents that suppress androgen catabolism, such as rofecoxib, have also had unexpectedly adverse effects on IHD (*Schooling, 2016*). Mechanisms by which androgens might cause IHD have not been extensively investigated, but likely involve coagulation and red blood cell attributes. Several haemostatic and thrombotic factors, such as thromboxane $A_2$ (*Pignatelli et al., 2012*; *Ajayi and Halushka, 2005*), endothelin-1 (*Polderman et al., 1993*; *van Kesteren et al., 1998*; *Sahebkar et al., 2015*), nitric oxide (*Pignatelli et al., 2012*; *Rosselli et al., 1998*) and possibly thrombin (*Orsi et al., 2019*; *Ferenchick et al., 1995*), may be driven by androgens and likely play a role in IHD (*Schooling et al., 2018b*; *Zhao, 2018*; *Nikpay et al., 2015*). Von Willebrand factor (*Sahebkar et al., 2016*) and

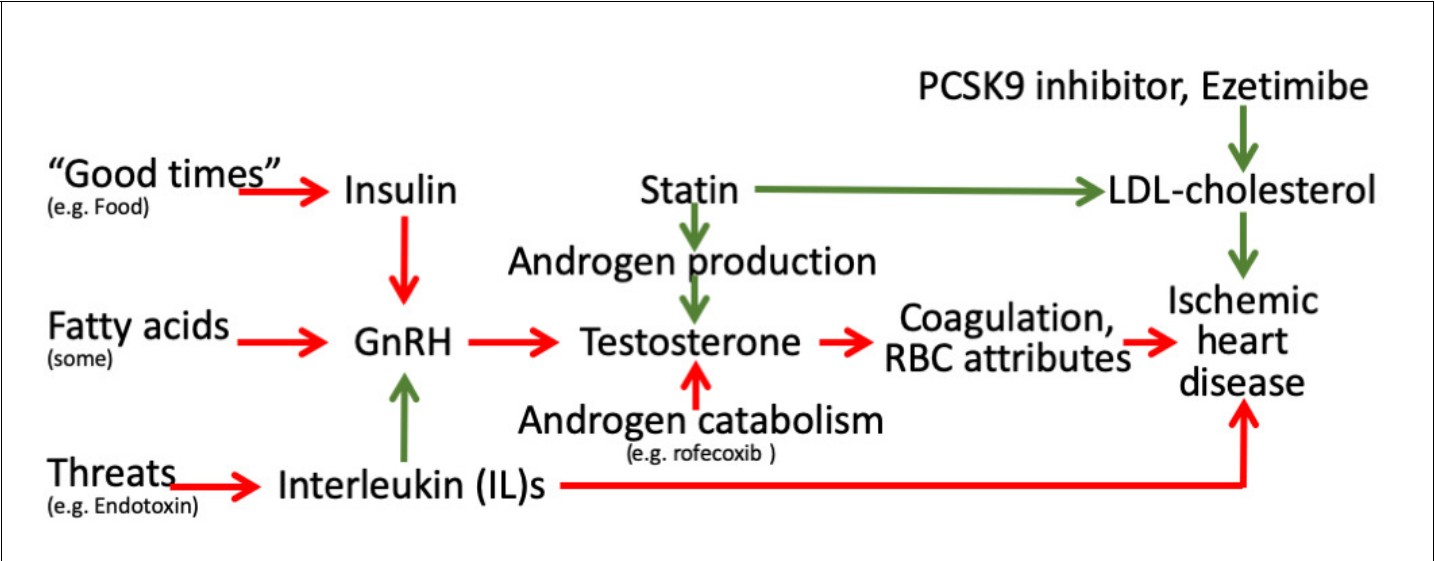

**Figure 2.** Schematic diagram showing the well-established protective effects of lipid modifiers on IHD (solid green lines) in the context of additional relevant pathways (green protective, red harmful) from an evolutionary biology perspective. (Key: GnRH: gonadotropin releasing hormone, RBC: red blood cell, LDL: low density lipoprotein).

asymmetric dimethylarginine (*Serban et al., 2015*) are also modulated by statins and may cause IHD (*Aday and Ridker, 2018*; *Au Yeung et al., 2016*), whether they are driven by testosterone is unknown. Several red blood cell attributes are affected by androgens, from reticulocytes to hematocrit (*Fernández-Balsells et al., 2010*; *Kanias et al., 2016*), but exactly which causes IHD is unclear, although reticulocytes are a possibility (*Astle et al., 2016*). Currently, comprehensive genetic validation of these pathways is hampered by the lack of availability of large sex-specific genome wide association studies (GWAS) of cytokines and coagulation factors.

Despite providing information that may be relevant to the performance of statins, and to the development of other therapies to protect against cardiovascular disease (*Schooling, 2017*), some limitations of this study exist. First, valid instruments should fulfill three assumptions, that is relate strongly to the exposure, not be associated with potential confounders and satisfy the exclusion restriction assumption. The F-statistics were >10. Despite high conditional F-statistics for testosterone the conditional F-statistics for the genetic mimics of statins were quite low and the Q-statistics for instrument validity were high suggesting pleiotropy, which we addressed by using multivariable MR-Egger. The associations with testosterone in women were not adjusted for factors, such as menopausal status, hormone use and history of oophorectomy, which could result in imprecision and weaker instruments. The SNPs used to mimic effects of statins, PCSK9 inhibitors and ezetimibe are well established (*Ference et al., 2019*), and in genes that harbor the target of each lipid modifier (*HMGCR*, *PCSK9* and *NCP1L1* respectively). We did not include body mass index (BMI) as a risk factor explaining the effect of statins on IHD, because statins increase BMI (*Swerdlow et al., 2015*) and decrease the risk of IHD, so including them in the multivariable analysis may inflate the effect of mimicking statins on IHD, rather than explaining part of their effect on IHD. The SNPs mimicking effects of the anti-inflammatory anakinra have been validated as increasing IL-1Ra (*Interleukin 1 Genetics Consortium, 2015*), and the SNP used to mimic effects of tocilizumab is well-established as affecting IL-6r (*Swerdlow et al., 2012*). Testosterone's effects on IHD in men could be via adiposity, insulin or LDL-cholesterol rather than via testosterone. However, consistent with a previous MR study, we found testosterone did not affect BMI in men (*Eriksson et al., 2017*), we also found little evidence that testosterone in men affected LDL-cholesterol (data not shown). We could not test whether testosterone in men affects insulin because of the lack of an insulin GWAS including the X chromosome. Sex-specific genetic associations were used throughout with exception of the genetic mimics of effects of anakinra and tocilizumab on IL1Ra and IL-6r respectively. However, inflammation operating on the reproductive axis would be expected to have sex-specific effects not sex-specific drivers. We selected between correlated SNPs based on p-values which is relatively arbitrary, and

the estimates could be sensitive to the choice of SNPs. Repeating the analysis using a larger number of correlated SNPs, where possible, taking into account their correlation, gave a similar interpretation. MR studies can be confounded by population stratification. However, we used genetic associations from GWAS mainly comprising people of white British ancestry with genomic control. Functions of each SNP predicting the exposures are not all fully understood, so we cannot rule out the possibility that the SNPs are linked with IHD through other pathways although we used sensitivity analysis.

We used SNPs predicting testosterone, but not other exposures, obtained from the same study as the genetic effects on IHD. However given the estimates for testosterone were largely obtained from non-cases, the overlap unlikely introduced substantial bias (*Burgess et al., 2016*). Canalization, that is buffering of genetic factors during development, may occur however; whether it does so is unknown. Our findings, largely in Europeans, may not be applicable to other populations. However, causes are unlikely to act differently in different populations, although the causal mechanisms may not be as relevant in all settings (*Lopez et al., 2019*). The SNPs mimicking effects of statins, PCSK9 inhibitors and ezetimibe were previously selected for their relations with LDL-cholesterol and on functional grounds (*Ference et al., 2019*), assuming the lipid modifiers act by action on lipids (*Ference et al., 2019*), so it is possible that relevant SNPs might have been discarded if they work through other mechanisms independent of lipids. It is also possible that the SNPs mimicking lipid modifiers might act via a different lipid trait, such as apoB (*Richardson et al., 2020*). The SNPs mimicking effects of anakinra and tocilizumab were similarly selected. Replication based on genetic instruments functionally relevant to all the exposures would be ideal. However, we used the most recent, published genetic instruments for testosterone (*Ruth et al., 2020*). Replication based on another large sex-specific IHD GWAS where the IHD cases are not from the same study as the testosterone instruments, would be ideal. However, sex-specific summary statistics are not available for large existing IHD GWAS, such as CARDIoGRAM (*Nikpay et al., 2015*). Moreover, the UK Biobank has the advantage of being very intensively genotyped and including the X chromosome, which is important for testosterone, but is not usually included in publicly available summary statistics. Lack of replication is a limitation of this study. Lastly, Mendelian randomization assesses the lifelong effects of an endogenous exposure rather than short-term effects of an interventions assessed in an RCT. Our estimates give an indication of the role of the exposures rather than the exact effects of the corresponding interventions. Nevertheless our estimates for statins on IHD are comparable with meta-analyses of statin trials considering similar outcomes (*Fulcher et al., 2015*).

Here, we present a hypothesis driven study examining the role of testosterone in mediating the effect of specifically statins in IHD, particularly in men. Future work could encompass a comprehensive sex-specific multivariable MR to confirm the role of sex hormones and sex hormone binding globulin in IHD as well as any mediation of their effects by key lipids, such as LDL-cholesterol or apoB. This work would be facilitated by the development of published genetic instruments for estrogen in women. Future work could also encompass assessing whether any other drugs that reduce cardiovascular disease, such as canakinumab (*Ridker et al., 2017*), also impact testosterone.

Taken together these complimentary findings for statins and anakinra raise the possibility that modulating testosterone, by whatever means, is a relevant feature for modulating IHD in men, with potential relevance to the development of new interventions, side-effects of existing interventions, re-purposing and appropriate use. Statins lowering testosterone could also be relevant to the muscle weakness or pain experienced on statins (*Collins et al., 2016*). Recognition that statins lower testosterone might also provide greater impetus for investigation of their role in other relevant conditions, such as prostate cancer (*Alfaqih et al., 2017*). Conversely, statins and anakinra did not clearly affect testosterone in women (*Table 1*) nor did testosterone mediate the effect of statins on IHD in women (*Table 3*). These differences by sex highlight the need for sex-specific approaches to IHD prevention and management, specifically in terms of the use of statins and investigation more broadly of causes of IHD.

## Conclusion

Genetic variants mimicking effects of statins and anakinra had opposite effects on testosterone and IHD in men, consistent with the effects of statins on IHD in men being partially mediated by testosterone. This insight that the pleiotropic effects of statins could be mediated by testosterone in men has implications for the use of existing interventions to prevent and treat IHD, the development of

new interventions for IHD and the re-use of statins for other androgen related conditions. Genetic confirmation that anakinra raises testosterone suggests its use in rheumatoid arthritis might have cardiovascular side-effects, particularly in men. It also highlights the importance of considering whether vulnerability to major diseases and interventions to promote lifespan need to be sex-specific.

## Materials and methods

### Genetic predictors mimicking effects of lipid and interleukin modifiers

Established genetic variants mimicking effects of statins, PCSK9 inhibitors and ezetimibe were taken from published sources (*Ference et al., 2019*) which selected SNPs from genes encoding proteins of the targets of each lipid modifier (*HMGCR* for statins, *PCSK9* for PCSK9 inhibitors and *NCP1L1* for ezetimibe) that lowered LDL-cholesterol. Genetic effects mimicking statins, PCSK9 inhibitors and ezetimibe were expressed in sex-specific effect sizes of LDL-cholesterol reduction taken from the largest available sex-specific GWAS summary statistics, that is the UK Biobank (http://www.nealelab.is/uk-biobank). The study was restricted to people of white British ancestry adjusted for the first 20 principal components, age, and age$^2$. In the main analysis for each lipid modifier, we only used independent ($r^2$ <0.05) SNPs most strongly associated with LDL-cholesterol. We obtained correlations between SNPs for each lipid modifier based on the 1000 Genomes catalog from LDlink (https://ldlink.nci.nih.gov). In sensitivity analysis, we used all the relevant SNPs for each lipid modifier, along with a matrix of their correlations. Established genetic variants mimicking effects of anakinra and tocilizumab and their effects on IL-1Ra and IL-6r respectively were also taken from published sources (*Swerdlow et al., 2012*; *Rafiq et al., 2007*).

### Sex-specific genetic predictors of testosterone

Strong (p-value<5 × 10$^{-8}$), independent ($r^2$ <0.05), sex-specific genetic predictors of testosterone were extracted from a published genome wide association study (GWAS) based on the UK Biobank and replicated in three independent studies (CHARGE Consortium, Twins UK and EPIC-Norfolk) (*Ruth et al., 2020*; *Sinnott-Armstrong et al., 2019*). Genetic associations with testosterone in this study were adjusted for genotyping chip/release of genetic data, age at baseline, fasting time and ten genetically derived principal components (*Ruth et al., 2020*). We used all 125 genetic variants given for bioavailable testosterone, hereafter testosterone, in men and all 254 genetic variants given for testosterone in women, as previously (*Zhao and Schooling, 2020*), because these had little correlation with sex hormone binding globulin (0.05 in men and 0.06 in women) (*Ruth et al., 2020*).

### Sex-specific genetic associations with IHD

Sex-specific genetic associations with IHD were taken from the UK Biobank individual data after excluding those with inconsistent self-reported and genotyped sex, excess relatedness (more than 10 putative third-degree relatives), abnormal sex chromosomes (such as XXY), or poor-quality genotyping (heterozygosity or missing rate >1.5%). The sex-specific associations with IHD obtained using logistic regression were adjusted for the first 20 principal components, age, and assay array. IHD was based on self-report at baseline, subsequent hospitalization diagnoses (primary or secondary) of International Classification of Diseases (ICD) 9 410–4 or ICD10 I20-5 and death registration causes (primary or secondary) of ICD10 I20-5 up until December 2019.

### Statistical analysis

The F-statistic was used to assess instrument strength, obtained using an approximation (mean of square of SNP-exposure association divided by square of its standard error) (*Bowden et al., 2016a*). A conventional threshold for the F-statistic is 10. SNPs with an F-statistic <10 were dropped.

Steiger filtering was used to check the directionality between testosterone and IHD. (*Hemani et al., 2017*) Sex-specific estimates of the associations of genetically predicted exposures (i.e., genetically mimicked effects of statins, PCSK9 inhibitors, ezetimibe, anakinra and tocilizumab) with testosterone and IHD, as well as estimates of the associations of genetically predicted testosterone with IHD were obtained by combining SNP-specific Wald estimates (SNP on outcome divided by SNP on exposure) using inverse variance weighting (IVW) with multiplicative random effects (*Burgess et al., 2013*). Multivariable MR was used to assess sex-specific associations of genetically

predicted exposures with IHD allowing for testosterone, accounting for correlations between SNPs on the same chromosome obtained from LDlink. In the multivariable MR, we pooled the genetic instruments mimicking statins and the genetic instruments for testosterone together, extracted their associations with LDL-cholesterol and testosterone and fitted one multivariable model. We estimated the Sanderson-Windmeijer multivariable conditional F-statistic (*Sanderson and Windmeijer, 2016*) to obtain a lower bound of the strength for each instrument conditional on the other exposure, and the Q statistics to asses pleiotropy, using the WSpiller/MVMR package (*Sanderson et al., 2019*). Given this analysis is multivariable by design with few genetic variants available to mimic the effects of statins, we used the multivariable MR-Egger estimates (*Rees et al., 2017*).

## Sensitivity analysis

Where possible we used methods with different assumptions to assess the validity of the univariable MR estimates from IVW, which assumes balanced pleiotropy. MR-Egger is valid as long as the instrument strength independent of direct effect assumption holds (*Bowden et al., 2015*). We also used a weighted median which gives valid estimates when more than 50% of information comes from valid SNPs (*Bowden et al., 2016b*). However, for exposures instrumented by correlated SNPs we did not give the weighted median or MR-Egger estimates because of concerns about their interpretability (*Burgess and Thompson, 2017*).

Given this is a hypothesis driven study, with a positive control, we used a statistical significance level of 0.05. All statistical analysis was conducted using R version 3.6.1 (The R Foundation for Statistical Computing, Vienna, Austria). The MendelianRandomization R package was used for the MR estimates. Estimates of genetic associations were taken from publicly available UK Biobank summary statistics, except the associations with IHD which were based on individual level genetic associations from the UK Biobank obtained under application #42468. All UK Biobank data were collected with fully informed consent.

## Acknowledgements

**Funding:** This research received no specific grant from any funding agency in the public, commercial or not-for-profit sectors.

## Additional information

### Funding
The authors declare that there was no funding for this work

### Author contributions
CM Schooling, Conceptualization, Formal analysis, Supervision, Validation, Methodology, Writing - original draft, Project administration; JV Zhao, Data curation, Validation, Writing - review and editing; SL Au Yeung, Resources, Data curation, Supervision, Investigation, Project administration, Writing - review and editing; GM Leung, Conceptualization, Methodology, Writing - review and editing

### Author ORCIDs
CM Schooling  https://orcid.org/0000-0001-9933-5887

### Ethics
Human subjects: This study is analysis of summary data previously collected with full consent.

### Decision letter and Author response
Decision letter https://doi.org/10.7554/eLife.58567.sa1
Author response https://doi.org/10.7554/eLife.58567.sa2

## Additional files

### Supplementary files

• Supplementary file 1. Associations of SNPs mimicking effects of lipid modifiers with LDL-cholesterol. (a) SNP-specific estimates for SNPs mimicking effects of statins, PCSK9 inhibitor and ezetimibe on LDL-cholesterol (effect size) in women and men from the UK Biobank, and for comparison estimates for both sexes together from the *Global Lipids* Genetics Consortium (GLGC) (*Willer et al., 2013*). (b) SNP-specific estimates for anakinra and tocilizumab SNPs on IL1-Ra (*Interleukin 1 Genetics Consortium, 2015*) and IL-6 (*Swerdlow et al., 2012*) respectively.

• Transparent reporting form

### Data availability

All data generated or analysed during this study are included in the manuscript and supporting files.

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
