## [Decision Letter]

**Acceptance summary:**

The authors performed a Mendelian randomization analysis of statin, PCSK9 inhibitor and ezetimibe use on ischemic heart disease to address a clinically relevant question – could the apparent additional benefits of statins and other similar treatments, over and above lowering lipids, be mediated by favourable effects on testosterone levels? They found a potentially interesting link between statins and testosterone – genetic variants in the gene encoding the HMGcoR receptor that are known to alter LDL-cholesterol levels.

**Decision letter after peer review:**

Thank you for submitting your article "Pleiotropic effects of statins on ischemic heart disease: a Mendelian Randomization study in the UK Biobank" for consideration by *eLife*. Your article has been reviewed by three peer reviewers, and the evaluation has been overseen by a Reviewing Editor and Matthias Barton as the Senior Editor. The following individual involved in review of your submission has agreed to reveal their identity: Timothy Frayling (Reviewer #3).

The reviewers have discussed the reviews with one another and the Reviewing Editor has drafted this decision to help you prepare a revised submission.

Title: Please modify the title to comply with *eLife* requirements. The title should not exceed 120 characters. Two part titles containing a colon punctuation mark (":") are not allowed.

Summary:

This is a neat study using genetics to ask a clinically relevant question – could the apparent additional benefits of statins and other similar treatments, over and above lowering lipids, be mediated by favourable effects on testosterone levels? The question becomes a little more complicated in that higher levels of testosterone tend to be beneficial for men's metabolic health but adverse for women and so the authors have used UK Biobank data, where it is easy to stratify analyses by sex.

The authors performed a Mendelian randomization (MR) analysis of statin, PCSK9 inhibitor and ezetimibe use on ischemic heart disease (IHD), and if these effects were mediated by testosterone in men or women using univariable and multivariable MR.

They have found a potentially interesting link between statins and testosterone – genetic variants in the gene encoding the HMGcoR receptor that are known to alter LDL-cholesterol levels (And so we know they are very good mimics for the on target effects of statins) and testosterone levels in men. The direction is that the alleles associated with lower LDL-cholesterol and therefore mimic statins also lower testosterone. There were no links for the other genes encoding the other targets of lipid lowering therapy. In addition, they assessed whether the genetic variants corresponding to anti-inflammatory effects of anakinra or tocilizumab use had effects on testosterone and IHD. The topic is very interesting.

1) The main finding is that, when adjusting for testosterone levels, the protective benefit of statins via the HMGcoR on heart disease attenuates from 0.54 odds ratio to 0.73 in men (when using the MR egger approach. There are wide 95% CIs around these estimates but the results imply that the testosterone effect is similar to the lowering LDL-cholesterol effect, which is extremely counter intuitive given that we know about LDL-cholesterol and heart disease. The reliance on a sensitivity analysis – MR egger – when the main analysis showed evidence of pleiotropy means the results could be very sensitive to slight differences in the parameters used. (SNPs and models).

2) A second concern is that the genetic variants in HMGCoR are associated with BMI and adiposity (see Swerdlow et al. Lancet publication using the HMGcoR SNPs to show an association with diabetes) whereas the others are less so. The authors need to assess the potential for the effect being mediated via BMI/adiposity rather than testosterone. Such a mechanism would be consistent with sex differences because women are at lower risk of heart disease than men for a given BMI. Likewise, many of the testosterone SNPs are highly pleiotropic, with many primarily associated with adiposity and insulin resistance and lipid levels. My concern is that the pleiotropy will lead to false inferences. The authors have partially addressed this with some approaches such as MR-Egger, but these are not infallible. The multivariable MR needs to include the adiposity measures. multivariable MR with adiposity measures, steiger filtering of SNPs that have larger effects on metabolic potentially mediating traits than they do on lipids or testosterone could be alternative approaches to try.

3) It would be relevant to see the results replicated in a two sample MR setting where the IHD cases are not from the same dataset as the testosterone and LDL SNPs. The heart disease GWAS consortia have not analysed separately by sex, which makes this very difficult, but there may be other large studies where this is possible ? If this cannot be done readily it should be discussed and noted as a limitation.

4) All of this means that the conclusion that "statins partially operate on IHD by reducing testosterone in men" is likely overstated.

5) A further major concern is the selection of SNPs in vicinity of the *HMGCR*, *NCP1L1* and *PCSK9* genes as instruments for the statin, PCSK9 and ezetimibe use, respectively. For example, a lookup of the instrument included for ezetimibe use (rs10260606) in the publicly available sex-combined GWAS on ezetimibe treatment (http://www.nealelab.is/uk-biobank) resulted in an association p-value of 0.02 being a rather weak instrument. Also in the referenced publication of Ference et al., 2019, that was used as basis for the instrument selection, these SNPs were (to my understanding) selected to assess potential drug targets, and not to reflect medication use.

6) Which statistical model was used to calculate the sex-specific genetic associations with IHD (I assume logistic regression)? Please provide the number of cases and controls per sex-stratum that were finally included in this association analysis also in the main text (not only in the Abstract).

Please provide the p-value level that was applied to declare significance of the results.

7) By conducting a sex-stratified analyses, have the authors considered the impact of collider bias. As if the genetic instruments are associated with sex and the outcome measures are also associated with sex, by stratifying based on a common cause of the exposure and outcome, a distorted/erroneous association may occur.

8) Please give more details about multivariate MR analysis. E.g. did you pool the genetic proxies for statin and testosterone all together and then extracted the associations of all these SNPs with LDL-cholesterol and testosterone and fitted it in one model?

Revisions expected in follow-up work:

1) As the authors aim to investigate sex-specific benefits of statin therapies, why only focus on testosterone. I think you should also look at 17β-oestradiol and SHBG (both available in UKBB), and conduct a multivariate MR to determine which would be the independent causal factor for IHD among the three (e.g. PMID: 32203549) and then forward the independent hormones/proteins in a multivariate model with LDL-cholesterol or apoB (drug targets of lipid lowering drugs) to determine whether these hormones would exert independent roles in contributing to IHD beyond the lipids. If this is the case, then it makes sense to further determine whether the genetic instruments of the drug targets show a sex-specific association with the sex hormones. This issue could be addressed in the Discussion.

---

## [Author Response]

Revisions for this paper:1) The main finding is that, when adjusting for testosterone levels, the protective benefit of statins via the HMGcoR on heart disease attenuates from 0.54 odds ratio to 0.73 in men (when using the MR egger approach. There are wide 95% CIs around these estimates but the results imply that the testosterone effect is similar to the lowering LDL-cholesterol effect, which is extremely counter intuitive given that we know about LDL-cholesterol and heart disease. The reliance on a sensitivity analysis – MR egger – when the main analysis showed evidence of pleiotropy means the results could be very sensitive to slight differences in the parameters used. (SNPs and models).

Please accept our apologies for a counter-intuitive finding. We are in no way trying to question the role of LDL-cholesterol in heart disease. We are instead trying to address two important health issues. First, the higher rates of heart disease in men than women,^1^ which are a factor contributing to shorter life expectancy in men than women. Second, the apparent pleiotropic effects of statins, because their explication might facilitate the discovery of much needed new treatments for cardiovascular disease.

As regards, the size of the estimates for statins via HMGcoR on ischemic heart disease (IHD). After allowing for testosterone, the estimates in men changed from being larger than the estimates in women to being very similar to the estimates in women, as expected. However, we agree that the magnitude of the estimates in Mendelian randomization (MR) is difficult to interpret because MR estimates capture lifetime effects rather than a possibly short-term exposure. We have clarified this important point to the Discussion, by making the following change:

From: “Lastly, Mendelian randomization assesses the lifelong effects of an endogenous exposure rather than short-term effects of an interventions assessed in an RCT, nevertheless our estimates for statins on IHD are comparable with meta-analyses of statin trials considering similar outcomes.^2^”

To: “Lastly, Mendelian randomization assesses the lifelong effects of an endogenous exposure rather than short-term effects of an interventions assessed in an RCT. Our estimates give an indication of the role of the exposures rather than the exact effects of the corresponding interventions. Nevertheless, our estimates for genetically mimicked statins on IHD are comparable with meta-analyses of statin trials considering similar outcomes.^2^ ”

Please accept our apologies for being unclear about the use of the MR-Egger estimate. For univariable mendelian randomization (MR), MR-Egger is a sensitivity analysis. For the multivariable MR, when the analysis is multivariable by design with few genetic variants specifically predicting the exposures, as here for statins, then the MR-Egger estimate is most likely to be reliable.^3^ It is also recommended to use the MR-Egger multivariable estimate when the multivariable MR-Egger intercept is significant,^3^ as here. We have added this point to the Materials and methods:

“Given this analysis is multivariable by design with few genetic variants available to mimic the effects of statins, we used the multivariable MR-Egger estimates^3^.”

We also amended the Results to be clearer by making the following change:

From: “Correspondingly, the multivariable MR-Egger intercepts were significant in men and women, so we used the MR-Egger estimates ^3^.”

To: “Correspondingly, the multivariable MR-Egger intercepts were significant in men and women.”

2) A second concern is that the genetic variants in HMGCoR are associated with BMI and adiposity (see Swerdlow et al. Lancet publication using the HMGcoR SNPs to show an association with diabetes) whereas the others are less so. The authors need to assess the potential for the effect being mediated via BMI/adiposity rather than testosterone. Such a mechanism would be consistent with sex differences because women are at lower risk of heart disease than men for a given BMI. Likewise, many of the testosterone SNPs are highly pleiotropic, with many primarily associated with adiposity and insulin resistance and lipid levels. My concern is that the pleiotropy will lead to false inferences. The authors have partially addressed this with some approaches such as MR-Egger, but these are not infallible. The multivariable MR needs to include the adiposity measures. multivariable MR with adiposity measures, steiger filtering of SNPs that have larger effects on metabolic potentially mediating traits than they do on lipids or testosterone could be alternative approaches to try.

Thank you very much indeed for raising these concerns as to whether the pleiotropic effects of statins are via BMI rather than testosterone. We have considered this possibility in detail.

First, we checked, as expected from Swerdlow,^4^ that statins increase BMI in men and women (beta 0.33 effect size, 95% confidence interval (CI) 0.20 to 0.47, in men and 0.31, 95% CI 0.22 to 0.40, in women, Author response table 1 below), using published instruments for BMI^5^ and rs12916-T to mimic effects of statins, and the UK Biobank sex-specific genetic associations with BMI (http://www.nealelab.is/uk-biobank).

Second, we considered the effect of BMI^5^ and genetically mimicked statins on IHD in univariable and multivariable MR. In univariable MR, as expected, BMI increased IHD (odds ratio (OR) 1.33, 95% confidence interval (CI) 1.15 to 1.55, in men and OR 1.23, 95% CI 1.02 to 1.48, in women) and genetically mimicked statins decreased IHD (OR 0.54, 95% CI 0.38 to 0.79 in men and had an estimate in the same direction in women OR 0.87, 95% CI 0.59 to 1.27, Author response table 1). In multivariable MR, considering the effects of BMI and genetically mimicked statins (rs12916-T) together on IHD, BMI had very similar associations with IHD as in the univariable analysis (OR 1.35, 95% CI 1.15 to 1.57 in men and women OR 1.27, 95% CI 1.07 to 1.52). In contrast, in the multivariable analysis including BMI genetically mimicked statins had a larger decreasing effect on IHD (OR 0.46, 95% CI 0.35 to 0.65 in men and OR 0.68, 95% CI 0.53 to 0.86 in women). As such, the protective effect of statins on IHD was increased by additionally considering BMI, so statins protective effect on IHD does not appear to be explained away by statins increasing BMI, in fact including BMI makes the protective effects of statins on IHD more marked. The multivariable MR is probably giving a larger effect size for statins than the univariable MR because it is showing the effects of “statins” after removing the harmful aspect driven by raising BMI. As such, statins raising BMI is unlikely to be a pathway by which statins protect against IHD, and we did not consider it further in the multivariable MR to keep the paper as simple and focused as possible. We have added this point in the Discussion as follows:

“We did not include body mass index (BMI) as a risk factor explaining the effect of mimicking statins on IHD, because statins increase BMI^4^ and decrease the risk of IHD, so including them in the multivariable analysis may inflate the effect of mimicking statins on IHD, rather than explaining part of the effect”.

We agree that the testosterone SNPs may have pleiotropic effects possibly acting via adiposity, insulin resistance or lipids. Downstream effects of the exposures help explain how the exposure affects the outcome. Downstream effects of the exposures generate vertical pleiotropy which does not bias MR estimates, because it is a pathway through the exposure, rather than the genetic instrument operating by a pathway distinct from the exposure. Pleiotropy biases MR estimates, when it is horizontal pleiotropy, and the genetic instrument affects the outcome by a pathway distinct from the exposure. We also checked whether the testosterone SNPs operated via adiposity or lipids. The testosterone SNPs did not affect BMI in men, consistent with a previous Mendelian randomization study,^6^ as given below (Author response table 2). In contrast testosterone raised BMI in women and reduced LDL-cholesterol, however, testosterone in women does not affect IHD. We have added that these points in the Discussion.

“Testosterone’s effects on IHD in men could be via adiposity, insulin or LDL-cholesterol rather than via testosterone. […] We could not test whether testosterone in men affects insulin because of the lack of an insulin GWAS including the X chromosome.”

Author response table 1: Univariable Mendelian randomization estimates for genetically mimicked statins (rs21916-T) and genetically predicted BMI (95 variants*) on IHD in men and women in the UK Biobank.

*after removing one variant predicting BMI (rs2112347) that was correlated with rs12916

Author response table 2: Univariable sex-specific Mendelian randomization estimates for testosterone on body mass index and LDL-cholesterol in men and women in the UK Biobank.

As suggested, we used Steiger filtering to check the direction from testosterone to IHD,^7^ which confirmed that the direction of causality was from testosterone to IHD in men and women. We have added this point in the Materials and methods and the Results.

In the Materials and methods:

“Steiger filtering was used to check the directionality between testosterone and IHD.^7^”

In the Results;

“Steiger filtering indicated directionality from testosterone to IHD in men and women.”

3) It would be relevant to see the results replicated in a two sample MR setting where the IHD cases are not from the same dataset as the testosterone and LDL SNPs. The heart disease GWAS consortia have not analysed separately by sex, which makes this very difficult, but there may be other large studies where this is possible ? If this cannot be done readily it should be discussed and noted as a limitation.

Please accept our apologies for being unclear. The testosterone SNPs were derived from the UK Biobank, but were also replicated in three independent studies,^8^ so they do have some replication. The LDL SNPs for statins, PCSK9 inhibitors and ezetimibe do not come from the same dataset as the IHD cases. The LDL SNPs come from Ference et al., as used previously to mimic the effects of lipid modifiers^9^. Only the beta co-efficients for the LDL SNPs on LDL come from the same study as the IHD cases. The beta coefficients are simply scaling factors which are quite similar to those from the Global Lipids Genetic consortium (GLGC), as we show in Supplementary file 1. We have added a column to that table to make it clearer. However, the GLGC does not have sex-specific genetic associations with LDL-c, so we used UK Biobank which gives sex-specific effect of the SNPs mimicking statins on LDL-c. We have explained this point, as given below:

“Strong (p-value<5×10^-8^), independent (r^2^<0.05), sex-specific genetic predictors of testosterone were extracted from a published genome wide association study (GWAS) based on the UK Biobank and replicated in three independent studies (CHARGE Consortium, Twins UK and EPIC-Norfolk) ^10 11^.”

“Genetic effects mimicking statins, PCSK9 inhibitors and ezetimibe were expressed in sex-specific effect sizes of LDL-cholesterol reduction taken from the largest available sex-specific GWAS summary statistics, i.e., the UK Biobank (http://www.nealelab.is/uk-biobank).”

Thank you for this suggestion about replication, we would be very keen to do so. We cannot identify any other sex-specific GWAS of IHD. Most GWAS of IHD are composed largely of men, so one option would be to repeat the analysis using an IHD GWAS of mainly men. However, there are two issues with that approach. First a GWAS of mainly men does not provide replication by sex. Second, almost all previous GWAS are autosomal, so we are missing key items in the testosterone instrument from the X chromosome, which means this is not a true replication. We have clarified this point in the Discussion, by making the following change:

From: “Replication based on another large IHD GWAS, such as CARDIoGRAM^12^, would be ideal, but it is less intensively genotyped than the UK Biobank and excludes the X chromosome, making it unsuitable.”

To: “Replication based on another large sex-specific IHD GWAS where the IHD cases are not from the same study as the testosterone instruments, would be ideal. However, sex-specific summary statistics are not available for large existing IHD GWAS, such as CARDIoGRAM.^12^ Moreover, the UK Biobank has the advantage of being very intensively genotyped and including the X chromosome, which is important for testosterone, but is not usually included in publicly available summary statistics. Lack of replication is a limitation of this study.”

4) All of this means that the conclusion that "statins partially operate on IHD by reducing testosterone in men" is likely overstated.

We fully accept that no study is ever definitive, so we have amended this sentence to say

“statins could partially operate on IHD by reducing testosterone in men".

5) A further major concern is the selection of SNPs in vicinity of the HMGCR, NCP1L1 and PCSK9 genes as instruments for the statin, PCSK9 and ezetimibe use, respectively. For example, a lookup of the instrument included for ezetimibe use (rs10260606) in the publicly available sex-combined GWAS on ezetimibe treatment (http://www.nealelab.is/uk-biobank) resulted in an association p-value of 0.02 being a rather weak instrument.

Thank you for these comments. Please accept our apologies for being unclear, we did not obtain the instruments for statins, PCSK9 or ezetimibe use from the UK Biobank GWAS of self-reports of drug use. There were three reasons for this choice. First, for comparability with other studies we used instruments from a previous study.^9^ Second, self-reports are not always very reliable, so a GWAS based on self-reports may generate instruments that represent noise rather than signal. This may be why MR studies exploring the effects of medications usually use genetic predictors related to the drug target, based on biological pathways so as to minimise potential bias. Third, the SNPs for statins, PCSK9 or ezetimibe are to some extent selected on functional rather than statistical grounds from relevant genes which encode proteins targeted by these drugs. This approach has the advantage of being functionally relevant, and so hopefully is relevant to the drugs biological effect, and shows the effect of the drug. The SNPs we used all had F-statistics greater than 10. On reflection, we realize that the language used to describe exposure was unclear as “effects of statin use” and we have amended it throughout, from “effects of statin use”, to “mimic the effect of statins” or similar. Thank you for pointing this out.

Also in the referenced publication of Ference et al., 2019, that was used as basis for the instrument selection, these SNPs were (to my understanding) selected to assess potential drug targets, and not to reflect medication use.

Thank you for these comments. As noted in the previous comment, we found it very tricky to find exactly the right words to describe the instruments for statins, PCSK9 inhibitors and ezetimibe clearly. We have carefully reviewed the underlying publications to find more appropriate wording to describe use of these variants in similar situations. As given below the original publications use the words “mimic the effect of [the drug] or “as a proxy for [drug] treatment”.

Ference et al., 2019, used the wording “Genetic variants that mimic the effect of …statins”^9^.

Ference et al., 2016, used the wording “We constructed genetic scores that mimic the effect of PCSK9 inhibitors and the effect of statins”^13^.

Ference et al., 2015, used the wording “To compare the biological effect of lower LDL-cholesterol mediated by inhibition of *NCP1L1*, *HMGCR*, or both on the risk of CHD, and to provide a context for interpreting the results of IMPROVE-IT, we sought to compare the effect of naturally random allocation to lower LDL-cholesterol on the risk of CHD mediated by genetic polymorphisms in the NPC1L1 gene (as a proxy for ezetimibe treatment), the *HMGCR* gene (as a proxy for statin treatment)”.

We have amended the paper throughout to be consistent with the previous usage and to use the expression “mimic the effect of” or similar throughout.

6) Which statistical model was used to calculate the sex-specific genetic associations with IHD (I assume logistic regression)? Please provide the number of cases and controls per sex-stratum that were finally included in this association analysis also in the main text (not only in the Abstract).

Thank you for pointing out this oversight, we have added that the associations with IHD were “obtained using logistic regression”. Information about the number of cases is given in the main text.

Please provide the p-value level that was applied to declare significance of the results.

We have added this point in the text:

“Given this is a hypothesis driven study, with a positive control, we used a statistical significance level of 0.05.”

7) By conducting a sex-stratified analyses, have the authors considered the impact of collider bias. As if the genetic instruments are associated with sex and the outcome measures are also associated with sex, by stratifying based on a common cause of the exposure and outcome, a distorted/erroneous association may occur.

Thank you very much indeed for raising the issue of collider bias. Stratifying on a common cause of exposure and outcome is typically used to address confounding, i.e., it removes bias rather than adds bias.^14^ Collider bias (or selection bias) typically occurs if there is selection (or stratification) on a common effect of the exposure and outcome,^14^ i.e., here the genetic variants and IHD. We do not think that sex is a result of the genetic variants or of IHD and hence collider bias is very unlikely upon stratification by sex.

8) Please give more details about multivariate MR analysis. E.g. did you pool the genetic proxies for statin and testosterone all together and then extracted the associations of all these SNPs with LDL-cholesterol and testosterone and fitted it in one model?

Please accept our apologies for not explaining these points clearly. This is exactly the process we carried out. We have added this point, as follows:

“In the multivariable MR, we pooled the genetic instruments mimicking statins and the genetic instruments for testosterone together, extracted their associations with LDL-cholesterol and testosterone and fitted one multivariable model.”

Revisions expected in follow-up work:1) As the authors aim to investigate sex-specific benefits of statin therapies, why only focus on testosterone. I think you should also look at 17β-oestradiol and SHBG (both available in UKBB), and conduct a multivariate MR to determine which would be the independent causal factor for IHD among the three (e.g. PMID: 32203549) and then forward the independent hormones/proteins in a multivariate model with LDL-cholesterol or apoB (drug targets of lipid lowering drugs) to determine whether these hormones would exert independent roles in contributing to IHD beyond the lipids. If this is the case, then it makes sense to further determine whether the genetic instruments of the drug targets show a sex-specific association with the sex hormones. This issue could be addressed in the Discussion.

Thank you very much indeed for asking why we focused on testosterone. We have for many years been examining the theory that statins operate specifically by testosterone,^15 16^ as presented explicitly in Figure 1 of the paper. In contrast, we have not been investigating the role of estrogen in IHD. Large randomized controlled trials have shown that hormone replacement therapy in women does not protect against IHD,^17^ and estrogen does not protect against IHD in men.^18^ Nevertheless, the possibility of a role of estrogen in IHD remains, because hormone replacement therapy may differ from naturally occurring estrogen in women and in men the Coronary Drug project estrogen arms were stopped early. Sex hormone binding globulin (SHBG) could be an independent influence on IHD or an important moderator of the role of testosterone, which deserves investigation. However, as shown in Author response table 3 (below) estrogen and SHBG do not seem relevant to specifically the effects of statins on IHD in men.

Author response table 3: Mendelian randomization estimates for effects of genetically mimicked statins (per effect sizes of LDL-cholesterol) on estrogen and SHBG in men and women from the UK Biobank summary statistics (http://www.nealelab.is/uk-biobank).

The unit of LDL-cholesterol is approximately 0.83mm/L.

We agree completely that it is a very important question to assess the independent role of testosterone, 17β-oestradiol and SHBG in IHD by sex and the extent to which they operate via LDL-cholesterol or apoB. Currently there is a dearth of comprehensive genetic instruments for estrogen. Ruth et al., by far the largest GWAS of sex hormones, only provided genetic instruments for estrogen in men, but not for estrogen in women.^19^ We have explained this future work in the Discussion as follows.

“Here, we present a hypothesis driven study examining the role of testosterone in mediating the effect of specifically statins in IHD, particularly in men. Future work could encompass a comprehensive sex-specific multivariable MR to confirm the role of sex hormones and sex hormone binding globulin in IHD as well as any mediation of their effects by key lipids, such as LDL-cholesterol or apoB. This work would be facilitated by the development of published genetic instruments for estrogen in women.

References:

https://wwwbiorxivorg/content/101101/660506v1

1) Moran AE, Tzong KY, Forouzanfar MH, et al. Variations in ischemic heart disease burden by age, country, and income: the Global Burden of Diseases, Injuries, and Risk Factors 2010 study. Glob Heart 2014;9(1):91-9. doi: 10.1016/j.gheart.2013.12.007 [published Online First: 2014/07/01]2) Fulcher J, O'Connell R, Voysey M, et al. Efficacy and safety of LDL-lowering therapy among men and women: meta-analysis of individual data from 174,000 participants in 27 randomised trials. Lancet (London, England) 2015;385(9976):1397-405. doi: 10.1016/s0140-6736(14)61368-4 [published Online First: 2015/01/13]3) Rees JMB, Wood AM, Burgess S. Extending the MR-Egger method for multivariable Mendelian randomization to correct for both measured and unmeasured pleiotropy. Stat Med 2017;36(29):4705-18. doi: 10.1002/sim.7492 [published Online First: 2017/09/30]4) Swerdlow DI, Preiss D, Kuchenbaecker KB, et al. HMG-coenzyme A reductase inhibition, type 2 diabetes, and bodyweight: evidence from genetic analysis and randomised trials. Lancet (London, England) 2015;385(9965):351-61. doi: 10.1016/s0140-6736(14)61183-1 [published Online First: 2014/09/30]5) Larsson SC, Bäck M, Rees JMB, et al. Body mass index and body composition in relation to 14 cardiovascular conditions in UK Biobank: a Mendelian randomization study. European heart journal 2020;41(2):221-26. doi: 10.1093/eurheartj/ehz388 [published Online First: 2019/06/14]6) Eriksson J, Haring R, Grarup N, et al. Causal relationship between obesity and serum testosterone status in men: A bi-directional mendelian randomization analysis. PLoS One 2017;12(4):e0176277. doi: 10.1371/journal.pone.0176277 [published Online First: 2017/04/28]7) Hemani G, Tilling K, Davey Smith G. Orienting the causal relationship between imprecisely measured traits using GWAS summary data. PLoS genetics 2017;13(11):e1007081. doi: 10.1371/journal.pgen.1007081 [published Online First: 2017/11/18]8) Rafiq S, Frayling TM, Murray A, et al. A common variant of the interleukin 6 receptor (IL-6r) gene increases IL-6r and IL-6 levels, without other inflammatory effects. Genes Immun 2007;8(7):552-9. doi: 10.1038/sj.gene.6364414 [published Online First: 2007/08/03]9) Ference BA, Ray KK, Catapano AL, et al. Mendelian Randomization Study of ACLY and Cardiovascular Disease. The New England journal of medicine 2019;380(11):1033-42. doi: 10.1056/NEJMoa1806747 [published Online First: 2019/03/14]10) Ruth KS, Day FR, Tyrrell J, et al. Using human genetics to understand the disease impacts of testosterone in men and women. Nat Med 2020;26(2):252-58. doi: 10.1038/s41591-020-0751-511) Sinnott-Armstrong N, Tanigawa Y, Amar D, et al. Genetics of 38 blood and urine biomarkers in the UK Biobank. 201912) Nikpay M, Goel A, Won HH, et al. A comprehensive 1,000 Genomes-based genome-wide association meta-analysis of coronary artery disease. Nature genetics 2015;47(10):1121-30. doi: 10.1038/ng.3396 [published Online First: 2015/09/08]13) Ference BA, Robinson JG, Brook RD, et al. Variation in PCSK9 and HMGCR and Risk of Cardiovascular Disease and Diabetes. The New England journal of medicine 2016;375(22):2144-53. doi: 10.1056/NEJMoa1604304 [published Online First: 2016/12/14]14) Hernán MA, Hernández-Díaz S, Robins JM. A structural approach to selection bias. Epidemiology 2004;15(5):615-25. doi: 10.1097/01.ede.0000135174.63482.43 [published Online First: 2004/08/17]15) Schooling CM, Au Yeung SL, Freeman G, et al. The effect of statins on testosterone in men and women, a systematic review and meta-analysis of randomized controlled trials. BMC Med 2013;11:57. doi: 10.1186/1741-7015-11-57 [published Online First: 2013/03/02]16) Schooling CM, Au Yeung SL, Leung GM. Why do statins reduce cardiovascular disease more than other lipid modulating therapies? Eur J Clin Invest 2014;44(11):1135-40. doi: 10.1111/eci.12342 [published Online First: 2014/09/25]17) Manson JE, Chlebowski RT, Stefanick ML, et al. Menopausal hormone therapy and health outcomes during the intervention and extended poststopping phases of the Women's Health Initiative randomized trials. Jama 2013;310(13):1353-68. doi: 10.1001/jama.2013.278040 [published Online First: 2013/10/03]18) The Coronary Drug Project. Findings leading to discontinuation of the 2.5-mg day estrogen group. The coronary Drug Project Research Group. Jama 1973;226(6):652-7. [published Online First: 1973/11/05]19) Willer CJ, Schmidt EM, Sengupta S, et al. Discovery and refinement of loci associated with lipid levels. Nature genetics 2013;45(11):1274-83. doi: 10.1038/ng.2797 [published Online First: 2013/10/08]et al.